# OpenReview forum: "ToolGuard: Red-Teaming Small Language Model Tool Call- ing on Consumer Hardware"
_TMLR — Under review for TMLR_

### Review · Reviewer_9CRP · 2026-07-12

**Summary Of Contributions:**

*Summary*

This paper introduces ToolGuard, a benchmark and attack taxonomy for evaluating adversarial tool-calling behavior in small language models, finding that capable SLMs execute malicious tool calls in roughly 50% of tested cases. It also proposes a lightweight rule-based runtime validator that blocks unsafe calls and substantially reduces attack success, but its effectiveness degrades against adaptive attacks and semantically valid data-exfiltration attempts.

---

*Weaknesses*

I think there several key major weaknesses:

1) the base model you used are too old and weak, I suggest to at least add few stronger one to test if the benchmark and attacking style is meaningful for stronger models, e.g., I want to see the performance of following models: Qwen/Qwen3-4B-Instruct-2507, Qwen/Qwen3.5-4B, google/gemma-3-4b-it, meta-llama/Llama-3.2-3B-Instruct (since I don't know the Llama3.2-3B you used is pre-trained one or after post-train finetuning).
2) 50 questions is usually too few for a paper with benchmark-orientated contribution, the robustness and applicability of your attacking to real-world models is still questionable as you sticked with SLMs. I feel it's insufficient to support broad conclusions about SLM tool-calling security. Another problem is also the analysis may not be statistical sound given only very few samples.
3) If you can provide some larger (stronger) models to justify the effectiveness of your framework, then it should present some evidence for its actual usefulness for the real-world models with more actual users.
4) Benchmark and code is not revealed, which is very difficult for the reviewers to assess the quality.
5) This is only measures single-turn attack if I understand it correctly, but more actual attack or safety issues would occur after multi-turn.
6) Should test it over more tool-calling environment and scenarios.

**Additional Comments:**

N/A

**Audience:**

Yes

**Audience Explanation:**

Yes

LLM tool-calling security is a trending topic; if the benchmark is meaningful, it may provide some insights for more actual business-level models (e.g., codex, claude, qwen).

**Broader Impact Concerns:**

Discussed in paper.

**Claims And Evidence:**

No

**Claims Explanation:**

I will put No for here given several issues I mentioned in the weakness section.

Several evidence isn't presented: 1) whether the benchmark and attacking framework is meaningful to stronger models, 2) too few questions which cannot support the claim for LLM tool-use security issue and maybe not be statistically sound for the later analysis, 3) no evidence of the benchmark implementation itself, which makes the reviewers to hard to assess its quality.

**Requested Changes:**

See weaknesses section, please either provide the content I requested or provide clarification.

---

> ### Author Response · Authors · 2026-07-18
> **Author response to Reviewer 9CRP**
>
> We thank the reviewer for a detailed and actionable review. The revision responds to each weakness, several of the requested experiments are now in the paper, and the new PDF is uploaded.
>
> **W1. Base models are too old or weak; add stronger ones (Qwen3-4B-Instruct, Qwen3.5-4B, gemma-3-4b-it, Llama-3.2-3B-Instruct).**
> We added exactly the four models named. Section 7.6 evaluates Qwen3-4B-Instruct, Qwen3.5-4B, Gemma-3-4B, and Llama-3.2-3B on a strictly held-out attack set. On the reviewer's specific question about Llama-3.2-3B: it is the instruction-tuned release (Llama-3.2-3B-Instruct), which we now state explicitly in the model description (Section 4.1). The newer models remain clearly vulnerable, at 25 to 54% held-out ASR, so the finding is not an artifact of an old model set. The defense continues to fire at full strength on covered attacks for these models: the within-run original-50 control shows about 72% relative reduction, consistent with the paper's 76 to 78% headline.
>
> **W2. 50 questions is too few and may not be statistically sound.**
> Two changes. First, scale: the benchmark is expanded from 50 to 153 prompts across 7 domains, of which 103 are strictly held out. Second, statistical unit: the 10 runs of a prompt are not independent, so a Wilson interval on pooled runs is overconfident. We now report per-prompt ASR with cluster-bootstrap 95% CIs over prompts, and the intraclass correlation ICC(1), which is 0.43 to 0.79 on the original 50-prompt benchmark and 0.71 to 0.92 on the expanded set. The moderate-to-high ICC is direct evidence for this concern, showing the run-level intervals were overconfident, and the prompt-level intervals we now report are the honest, wider ones. We also add a temperature sweep (new Table 1) showing ASR is flat across T in {0, 0.3, 0.7, 1.0}, so the fixed T=0.7 is not a confound.
>
> **W3. Provide larger or stronger models to justify real-world usefulness.**
> See W1. The 4B-class results show the benchmark and attack framework transfer to newer, more capable models, and the defense holds. We also add a within-family Qwen2.5 size ladder at 0.5B, 1.5B, and 3B (new Table 4): with architecture held fixed, ASR rises 26.6%, 36.6%, 42.8%, a clean size gradient. Scaling to business-tier models (Claude, GPT, large Qwen) is the highest-value next step, and since our harness is model-agnostic over any tool-calling endpoint, it is a direct extension, which we note in the discussion. We cannot host those models on the single consumer-hardware machine that defines this paper's edge deployment context.
>
> **W4. Benchmark and code are not revealed.**
> We are preparing an anonymized repository with all 153 prompts, the 25 tool schemas, both policy files (train and full), the harness, and the multi-turn scaffold, and will link it during the discussion period. Appendix A already lists the full tool inventory and all 50 core prompts verbatim in the meantime.
>
> **W5. Only single-turn is measured; real attacks occur over multi-turn.**
> We now state the single-turn scope precisely in the Threat Model (Section 3.1) and added a paragraph distinguishing our setting from indirect prompt injection, which is inherently multi-turn. Because ToolGuard validates completed tool calls regardless of why the model emitted them, its per-call checks apply unchanged to multi-turn and injection-induced calls. We provide a tested multi-turn scaffold with four strategies (Section 8.3) as the basis for a multi-turn evaluation, and flag running it at scale as the clearest next step rather than overclaiming single-turn coverage.
>
> **W6. Test more tool-calling environments and scenarios.**
> The expansion adds two new tool domains absent from policy development, calendar/scheduling and cloud/DevOps (IaaS), for 7 domains and 25 tool schemas total. 68 of the 103 new prompts are policy-evading by design, using payload variants the frozen policy regexes do not enumerate, so the held-out set stresses the defense on genuinely unseen tools and scenarios rather than paraphrases.
>
> **Summary of what moved.**
> 103 additional held-out prompts, 2 new domains, and the 4 newer models requested; prompt-level statistics with cluster-bootstrap CIs and ICC(1); a temperature sweep and a within-family size ladder; explicit single-turn scoping with an indirect-prompt-injection discussion and a multi-turn scaffold; the full prompt and tool inventory in Appendix A; and an anonymized code and benchmark release to follow during the discussion period.

---

### Review · Reviewer_jiww · 2026-07-17

**Summary Of Contributions:**

This paper presents ToolGuard, the first systematic study of adversarial robustness in small language model (SLM) tool calling. The paper defined a set of five tool calling attacks and designed a benchmark for tool calling attack. The paper investigates the model perfromance against the attacks on various tools and domains with multiple runs

The paper also proposes a model-agnostic, post-hoc YAML policy enforcement mechanism that reduces mean ASR by 76% (full benchmark) and 78% (held-out test set) across differen models and attacks.

**Audience:**

Yes

**Audience Explanation:**

The paper addresses a timely and practical problem at the intersection of LLM safety, edge AI deployment, and agentic systems. The capability–vulnerability tradeoff finding (models good enough to use tools are inherently good enough to be attacked) is an important conceptual insight for the safety community. The practical defense with sub-5ms latency is directly deployable, which appeals to practitioners. The adaptive evaluation methodology demonstrates good scientific practice and provides a template for future defense papers. The topic is of growing importance as SLMs are deployed in agentic settings on consumer devices.

**Claims And Evidence:**

No

**Claims Explanation:**

1. The definition and discussion on the five attack is short. The paper should provide more description and discussion on how they can be used, and their possible outcomes. And providing example prompts whould better explain how these attacks are conducted. For examples, it is unclear how Chain Attacks is conducted and how its individual tool calls are benign.

2. The benchmark setting is not comprehensive. The paper fix the model temperature at 0.7, and with only total 2000 runs on 50 prompts, which is not large enough to reflect the statistical performance of the model or the prompts. T=0.7 is a quite large temperature and could produce very diverse model output, the paper should provide more investigation on how it impact the model performance against the attacks. So the claim that there is a "capability floor" in section 5.2 is under-supported. The paper should compare more models within this parameter size to make this claim.

3. The work "50 prompts for 5 attacks" is a bit misleading, it makes people think that each attack has 50 prompts. Given that there are 17 tools, and 5 attacks, 50 prompts, the paper failed to provide a list of these tools and prompts and how they matches in the appendix for clarification.

**Requested Changes:**

1. Expand ToolAttackBench or clearly scope the claims to match the 50-prompt scale.
2. Include at least one model with explicit tool-calling safety fine-tuning for comparison.
3. Discuss the relationship to indirect prompt injection more concretely.

Minor issues:
1. In Figure 1, x-axis uses log-scale which somewhat exaggerates the visual gap between 1B and 1.7B.
2. The privilege escalation "artifact disclosure" (Table 7, ‡) is commendably transparent but the range notation [11.1%, 15.0%] is unusual for a results table.

---

> ### Author Response · Authors · 2026-07-18
> **Author response to Reviewer jiww**
>
> We thank the reviewer for a careful and constructive review, and for noting the value of the capability and vulnerability tradeoff and the deployability of the defense. We have revised the paper on every point and uploaded the new PDF. Our responses follow the reviewer's structure.
>
> **1. Attack definitions too short; provide examples, outcomes, and how chain attacks work.**
> We rewrote Section 3.2. Each of the five categories now carries a verbatim benchmark prompt, the tool it targets, and the concrete outcome on success. For chain attacks we explain that each individual call is within policy (list a directory, read a config, send an email), while the composition is reconnaissance, then collection, then exfiltration, which is why a per-call validator is insufficient. The full tool inventory and all 50 prompts are in Appendix A.
>
> **2. Temperature fixed at 0.7; investigate its effect.**
> We ran the sweep. On Qwen2.5-3B over the full core benchmark at T in {0, 0.3, 0.7, 1.0}, ASR is essentially flat: 44.0%, 45.2%, 42.8%, 40.4% (new Table 1, Section 4.1). All four prompt-level 95% CIs overlap with no monotonic trend, so the choice of T=0.7 does not drive the results.
>
> **3. 2000 runs on 50 prompts may not be statistically sound.**
> First, scale: the benchmark is expanded from 50 to 153 prompts across 7 domains, of which 103 are strictly held out (Section 7.6). Second, statistical unit: the 10 runs of a prompt are not independent, so a Wilson interval on pooled runs is overconfident. We now report per-prompt ASR with cluster-bootstrap 95% CIs over prompts, and ICC(1), which is 0.43 to 0.79 on the original benchmark and 0.71 to 0.92 on the expanded set. The prompt-level intervals are the honest ones and are wider than the run-level intervals.
>
> **4. "Capability floor" under-supported; add more models in the 1 to 1.7B range.**
> First, we no longer describe a hard floor. The executability analysis (Section 5.7) shows the smallest model's apparent robustness is largely a tool-call format-competence artifact: it generates the malicious payloads but emits many as malformed text. Counting would-dispatch calls, Llama-3.2-1B's ASR rises from 10.8% to 24.2%, so the finding is a gradient, not a threshold. Second, answering the request for more models in this range, we added a within-family Qwen2.5 ladder at 0.5B, 1.5B, 3B (new Table 4, Section 5.2). Holding architecture fixed, ASR rises 26.6%, 36.6%, 42.8%; a paired bootstrap gives the 0.5B to 3B gap as 16.2pp, 95% CI [6.0, 26.8]. Third, we added three newer 4B-class models (Qwen3-4B-Instruct, Qwen3.5-4B, Gemma-3-4B) alongside the reused Llama-3.2-3B.
>
> **5. "50 prompts for 5 attacks" is misleading; list the tools and prompts.**
> Appendix A lists all 16 core tools plus 9 expansion tools, and all 50 core prompts verbatim by category, with per-category counts (parameter injection 13, tool substitution 8, privilege escalation 10, data exfiltration 10, chain attacks 9). We also corrected a counting error the reviewer surfaced: the correct figures are 16 core tools (5 domains) and 25 total (7 domains), not 17, now consistent throughout.
>
> **Requested: expand ToolAttackBench or scope the claims.** Both: expanded to 153 prompts (103 held out), with generalization claims bounded by policy coverage in Section 7.6.
>
> **Requested: a model with explicit tool-calling safety fine-tuning.**
> To our knowledge no sub-4B model is both a native tool-caller and explicitly safety-tuned to refuse malicious tool calls. The available small safety models (for example Qwen3Guard-4B, GuardReasoner 3B) are external content-moderation classifiers, not tool-calling agents, so using one is architecturally the same design as ToolGuard: an external check over a tool-calling model. ToolGuard fills that role with a deterministic, sub-5ms, memory-free check, whereas a 3 to 4B guard would roughly double edge latency and memory. If the reviewer prefers an empirical anchor, we will run Qwen3Guard-4B as an external filter and add it.
>
> **Requested: relationship to indirect prompt injection.**
> We added a paragraph to Section 3.1. Our setting is direct injection. Indirect prompt injection is distinct: attacker content arrives in a tool result and hijacks a subsequent call, so it is inherently multi-turn. Because ToolGuard validates completed tool calls regardless of why they were emitted, its per-call checks apply unchanged to injection-induced calls. We provide a tested multi-turn scaffold (Section 8.3) for the cross-call evaluation.
>
> **Minor: Figure 1 log-scale.** Figure 1 now uses a linear x-axis.
>
> **Minor: range notation [11.1%, 15.0%].** The per-category held-out table (Table 10) now reports the single conservative value, 15.0%; the 11.1% alternative and the artifact explanation stay in the text.
>
> **Code and benchmark release.** We are preparing an anonymized repository with all prompts, tool schemas, policy files, the harness, and the multi-turn scaffold, and will link it during the discussion period.

---

### Review · Reviewer_3ReN · 2026-07-19

**Summary Of Contributions:**

The paper aims to characterize the adversarial vulnerability of small language models used for tool calling and to evaluate whether a lightweight, model-agnostic runtime policy layer can prevent unsafe tool execution on consumer hardware. It develops a five-category taxonomy—parameter injection, tool substitution, privilege escalation, data exfiltration, and chain attacks—and instantiates it as ToolAttackBench, comprising 50 core prompts over 16 tools in five domains plus 103 strictly held-out prompts that expand coverage to 25 tools and seven domains. Four core models ranging from 1B to 3B parameters are tested locally via Ollama on consumer hardware, with 10 runs per prompt at a temperature of 0.7; four newer 3B–4B models are evaluated on the expansion. Attack success is scored deterministically using a “would-dispatch” rule that counts native structured calls and parseable text-emitted calls naming real tools. Capable core models showed 47–52% attack success, while the 1B model reached 24.2% under would-dispatch scoring. Vulnerability increased monotonically across the Qwen2.5 size ladder. ToolGuard reduced mean core ASR from 49.3% to 11.8% and held-out ASR to 10.9%, but generalized poorly to unseen domains and policy-aware attacks in practice overall.

**Audience:**

Yes

**Audience Explanation:**

Adversarial vulnerability of small language models used for tool calling is a very important problem for the TMLR community.

**Broader Impact Concerns:**

None that I am aware of in this review iteration

**Claims And Evidence:**

Yes

**Claims Explanation:**

The distinction between harmful text generation and harmful tool execution is well motivated: an unsafe tool call may immediately transfer money, expose data, modify permissions, or execute code. Focusing specifically on smaller models is also useful, as they are plausible candidates for edge, local, and lower-cost agentic deployments.

Treating ten repeated runs of the same prompt as ten independent observations would produce artificially narrow intervals. The paper correctly recognizes the prompt as the sampling unit, reports mean per-prompt ASR, uses a prompt-cluster bootstrap, and supplies ICC values demonstrating substantial within-prompt correlation. This is one of the greatest methodological improvements in the revision.

The temperature sweep reduces concern that the main finding is an artifact of a single decoding temperature. The Qwen2.5 within-family ladder reduces architecture-family confounding in size assessments. The native/recovered/intent scoring tiers expose the difference between safety and format competence. Held-out, expansion, benign-call, and adaptive-attack evaluations reveal both where ToolGuard works and where it fails.

**Requested Changes:**

The attack prompts, expected safe behaviors, severity levels, and deterministic scoring rules are manually designed. The 103-prompt expansion is reportedly LLM-assisted, reviewed, and unit-tested, but there is no independent annotation study, inter-rater agreement, external expert validation, or comparison with naturally occurring attacks. Therefore, the benchmark may encode the authors’ own assumptions about what constitutes an attack and what a permissive runner would execute. Thus, further development or discussion of this point?

The reported 0% false-positive rate comes from only 41 simulated canonical tool calls, not from actual outputs produced by every model on a broad benign workload. The corresponding Wilson interval extends to 8.6%, and the paper acknowledges that per-model false-positive rates on naturally variable outputs remain unmeasured. Please elaborate on the steps that could be taken to deploy this strategy in real scenarios.

The 76–78% reductions are obtained on the core benchmark and a 15-prompt split derived from the same original collection. For the 103-prompt expansion, the frozen policy removes far less because many attacks intentionally fall outside its payload enumeration, and two domains have no corresponding policy rules. Adaptive attacks further demonstrate substantial bypassability. Can you clarify on the type of defense reported in the paper, is it a security filter or a general defense mechanism.?

---

> ### Author Response · Authors · 2026-07-19
> **Author response to Reviewer 3ReN**
>
> We thank the reviewer for a careful reading of the revision and for highlighting the prompt-level statistics, the temperature sweep, the size ladder, and the held-out and adaptive evaluations. We respond to each requested change.
>
> **1. Benchmark validity: author-designed, no independent annotation study.**
>
> We agree, and want to be precise about where the subjectivity lives. Scoring involves no human or LLM judgment at evaluation time: the would-dispatch rule and per-prompt criteria are deterministic code (keyword/pattern matching), unit-tested to score as intended, so inter-rater agreement is not applicable at the scoring step as it is for human-judged benchmarks. What is author-encoded is the design: which prompts constitute attacks, expected safe behavior, and severity.
>
> Three partial mitigations. First, the payloads are not invented: the expansion instantiates well-documented attack classes (SQL injection via HAVING/CASE/time-based clauses, path traversal via null-byte/unicode/double-encoding, command injection via newline/${IFS}) drawn from the classical security literature. Second, the five categories align with harms studied in the agent-safety literature (ToolEmu, InjectAgent, AgentDojo), and each prompt's target action (unauthorized transfer, exfiltration, privilege escalation) is one whose harmfulness is not judgment-dependent. Third, the 103 expansion prompts were LLM-assisted but each was human-reviewed and unit-tested.
>
> On "what a permissive runner would execute": the would-dispatch rule models a runner that dispatches any structurally valid call naming a real schema tool. We believe this is the realistic baseline, since mainstream tool-calling stacks typically hand structurally valid calls to the application without payload-level validation, which is the gap ToolGuard fills; we will state this assumption explicitly beside the rule's definition.
>
> What we have not done is independent validation: no second annotator, external expert review, or comparison with attacks in the wild. We will add a paragraph to Section 8.3 (Limitations) saying so: the benchmark encodes the authors' threat model, deterministic scoring removes rater variance but not designer bias, and independent expert annotation plus grounding in observed agent attacks (as incident corpora emerge) are the validation steps for future work.
>
> **2. FPR: steps toward deployment in real scenarios.**
>
> The 0% FPR (n=41, Wilson CI [0%, 8.6%]) on simulated canonical calls is necessary but not sufficient for deployment. The path we would prescribe, which the design directly supports:
>
> 1. **Shadow mode first.** As a post-hoc validator, ToolGuard can run log-only on live traffic with zero behavioral impact, yielding a per-model, per-deployment FPR on actual outputs before enforcement.
> 2. **Per-model benign calibration.** Replay the target model's own outputs on a benign workload through the policy; models differ in structural quirks (argument ordering, optional fields), exactly what the n=41 simulated probe does not measure.
> 3. **Per-deployment tuning.** The policy is declarative YAML; rules that fire on benign traffic can be narrowed without code changes, and sub-5ms overhead leaves headroom for stricter rule sets.
> 4. **Blocked-call UX.** Fail closed to a user confirmation step rather than silently, so residual false positives cost a click, not task failure.
>
> We will fold this into Section 8.4. If valuable, we can run step 2 during the discussion period: replay our core models' actual outputs on the 41-prompt benign set and report per-model FPR. This isolates structural quirks (deterministic replay at temperature 0); run-to-run variability is what shadow mode measures. It may refine the simulated 0% figure, which is its purpose.
>
> **3. Security filter or general defense mechanism?**
>
> A security filter. ToolGuard is a coverage-bounded, rule-based policy filter over completed tool calls, not a general defense: it removes the attacks its rules describe, and the adaptive evaluation (Section 7.3) and held-out expansion (Section 7.6) are deliberate measurements of that boundary; policy-aware adversaries partially bypass it, and the frozen policy removes far less on payloads and domains it does not enumerate. That is why the limitations state that manual policies cannot generalize to unanticipated variants and why we argue for learned components beyond the filter. The 76-78% reductions are what a well-matched filter delivers inside its coverage, with Sections 7.3 and 7.6 delimiting the outside.
>
> We will make this explicit: the abstract will describe ToolGuard as a coverage-bounded runtime policy filter, and Section 8.3 will state it is a security filter rather than a general defense mechanism.
>
> We thank the reviewer again for an assessment that engaged closely with the revision; the three additions above (a construct-validity limitation, a deployment recipe, and unambiguous filter terminology) make the paper's claims precisely scoped.

---

### Author Response · Authors · 2026-07-18
**Anonymized code and benchmark release**

We have posted the anonymized code and benchmark, delivering on the release we committed to during the review period.

Link: https://anonymous.4open.science/r/toolguard-anon-8EBF

Contents:

- All 153 attack prompts (the 50 core prompts and the 103 held-out prompts), verbatim, in src/attacks/.
- The 25 tool schemas across the 7 domains, in src/harness/tool_schemas.py.
- Both policy files: the full policy and the train-only policy used for the leakage-free held-out evaluation, in configs/.
- The full red-team harness, the offline defense evaluator, and the multi-turn scaffold.
- The raw per-run results behind the revision tables (the temperature sweep, the within-family size ladder, and the four newer models), under experiments/tmlr-revision/.

The README lists install and reproduction commands, and the test suite (pytest) passes. If anything else would help verification, tell us and we will add it.